

# Three new *Penicillium* species isolated from the tidal flats of China

Ke-Xin Xu[1,*], Xia-Nan Shan[2,*], Yongming Ruan[2], JianXin Deng[1] and Long Wang[3]

[1] College of Agriculture, Yangtze University, Jingzhou, Hubei, China
[2] College of Chemistry and Life Sciences, Zhejiang Normal University, Jinhua, Zhejiang, China
[3] State Key Laboratory of Mycology, Institute of Microbiology, Chinese Academy of Sciences, Beijing, China
[*] These authors contributed equally to this work.

## ABSTRACT

During a survey of culturable fungi in the coastal areas of China, three new species of *Penicillium* sect. *Lanata-Divaricata* were discovered and studied with a polyphasic taxonomic approach, and then named as *P. donggangicum* sp. nov. (ex-type AS3.15900[T] = LN5H1-4), *P. hepuense* sp. nov. (ex-type AS3.16039[T] = TT2-4X3, AS3.16040 = TT2-6X3) and *P. jiaozhouwanicum* sp. nov. (ex-type AS3.16038[T] = 0801H2-2, AS3.16207 = ZZ2-9-3). In morphology, *P. donggangicum* is unique in showing light yellow sclerotia and mycelium, sparse sporulation, restricted growth at 37 °C, irregular conidiophores, intercalary phialides and metulae, and pyriform to subspherical conidia. *P. hepuense* is distinguished by the fast growth on CYA and YES and slow growth on MEA at 25 °C, weak or absence of growth at 37 °C, biverticillate and monoverticillate penicilli, and ellipsoidal conidia. *P. jiaozhouwanicum* is characterized by abundant grayish-green conidia *en masse* and moderate growth at 37 °C, the appressed biverticillate penicilli and fusiform, smooth-walled conidia. These three novelties were further confirmed by the phylogenetic analyses based on either the combined *BenA-CaM-Rpb2* or the individual *BenA*, *CaM*, *Rpb2* and internal transcribed spacer (ITS) sequences.

## INTRODUCTION

With the application of molecular approaches in fungal taxonomy and environmental studies, the number of the estimated fungal species on the earth may be more than doubled, *i.e.,* from about 1.5 million to 2.2–3.8 million. However, until now, the number of discovered species accounts for only 3–8%, or approximately 120,000–150,000 (*Hawksworth & Lücking, 2017*; *Hawksworth & Lücking, 2018*; *Lücking et al., 2020*; *Lücking et al., 2021*). The less-studied environments, for example, the coastal tidal flats, are thought to be the treasure house of new fungal species. China has a coastline of ca. 32,000 km and the tidal flat areas cover about 20,000 sq. km., which are potential valuable land resources either economically or ecologically (*e.g.,* *Long et al., 2016*; *Nicoletti & Trincone, 2016*).

Numerous studies on the fungal communities of coastal environments have been carried out based on culture-dependent and/or culture-independent approaches (*e.g.,* rDNA-based NGS) (*e.g.,* *Singh et al., 2012*; *Zhang et al., 2012*; *Heo et al., 2019*). For example,

Corresponding authors
JianXin Deng, djxin555@hotmail.com
Long Wang, wl_dgk@sina.com

*Li et al. (2016)* conducted an extensive study on the mycobiota of the intertidal regions along the coastline of China using an environmental metabarcoding approach with ITS2 as the barcode, which revealed the diverse fungal communities in mudflats, in which the facultative marine fungal taxa were in the majority, particularly, the members of some ubiquitous genera, such as *Penicillium*, *Aspergillus*, *Cladosporium*, *Fusarium* (as *Gibberella*) were abundant in aquaculture areas. However, the ITS-based metabarcoding approach alone cannot perform the accurate species identification for most of the fungal genera, especially *Penicillium*, *Aspergillus* and *Talaromyces* (*e.g.*, *Visagie et al., 2014a*; *Visagie et al., 2014b*; *Lücking et al., 2020*; *Houbraken, Visagie & Frisvad, 2021*). Thus, the integration of morphological, multi-locus phylogenetic and metabarcoding approaches is considered as a proper practice in studying environmental mycobiota (*e.g.*, *Lücking et al., 2020*). For instance, using morphological and multi-locus phylogenetic approaches based on ITS, *BenA* and *RPB2* sequences, *Park et al. (2014)* isolated 36 *Penicillium* taxa along the southern and eastern coastlines of Korea, among which nine were undetermined species. Employing the culture-dependent and culture-independent approaches with ITS-based and *BenA*-based metabarcoding, *Park et al. (2019)* discovered 96 taxa of *Penicillium* along the western and southern intertidal zones of Korea, among which 65 were isolated with a culture-dependent method, and eight might be new species.

The number of reported *Penicillium* species has more than doubled since 2000, namely, about 483 until now (*Pitt, Samson & Frisvad, 2000*; *Visagie et al., 2014a*; *Visagie et al., 2014b*; *Houbraken et al., 2020*). The understudied geographic areas and ecological habitats are considered as one major source for undiscovered species (*Hawksworth & Lücking, 2017*; *Hawksworth & Lücking, 2018*). With the extensive employment of molecular approaches in the exploration of these new ecological habitats or geographical areas, especially the environmental metabarcoding technique which has shown the powerful capability to uncover new taxa, many more members of this ubiquitous genus would be expected to be discovered, though a portion of OTUs may not be true natural taxa (*Lücking et al., 2020*; *Lücking et al., 2021*).

In a survey of culturable fungi in the tidal flats of China with a culture-dependent method, we isolated numerous strains of *Penicilium* and here propose three new species of sect. *Lanata-Divaricata* in line with the taxonomic scheme of *Houbraken et al. (2020)*, namely, *P. donggangicum* sp. nov., *P. hepuense* sp. nov., and *P. jiaozhouwanicum* sp. nov.

## MATERIALS & METHODS

### Isolation of strains

Soil samples were collected from the coastal mudflats of Yalu River Estuary Wetland Park, Donggang City, Liaoning (39°29′24″N, 124°19′12″E, 39 m); Dangjiang Town, Hepu County, Beihai City, Guangxi (21°18′13.68″N, 109°23′44.88″E, 10 m); Shatian Town, Babu District, Hezhou City, Guangxi (21°30′38″N, 109°39′58″E, 200 m); Jiaozhouwan, Qingdao City, Shandong (37°10′13″N, 120°7′36″E, 3 m); Jiulongkou, Zhangzhou City, Fujian (24°25′57N, 117°47′28E, 32 m). The dilution method of *Malloch (1981)* was used to isolate the culturable fungi with 0.1% agar water solution instead of water. Five distinctive strains

were isolated and deposited in China General Microbiological Culture Collection Center (CGMCC) as AS3.15900 = LN5H1-4 (*P. donggangicum* sp. nov.); AS3.16039 = TT2-4X3, AS3.16040 = TT2-6X3 (*P. hepuense* sp. nov.); AS3.16038 = 0801H2-2, AS3.16207 = ZZ2-3-9 (*P. jiaozhouwanicum* sp. nov.).

## Morphological studies

Colony characters were examined according to the methods and culturing media proposed by *Raper & Thom* (*1949*; Czapek agar, Cz), *Pitt* (*1979*; Czapek yeast autolysate agar, CYA, yeast extract; Oxoid, Hants, UK), *Samson et al.* (*2010*; 5% malt extract agar, MEA, malt extract; Oxoid, Hants, UK), and *Frisvad* (*1981*; yeast extract sucrose agar, YES, yeast extract; Oxoid, Hants, UK; creatine sucrose agar, CREA). The names of colors were referenced to *Kornerup & Wanscher (1978)*. Wet mounts were made by picking fungal materials with reproductive structures from a colony on MEA and teasing them apart with two needles in a small drop of 85% lactic acid water solution without dye. Microscopic observation and photographs were made with an Axioplan2 imaging and Axiophot2 universal microscope (Carl Zeiss, Oberkochen, Germany).

## Molecular studies

Genomic DNA was extracted with a Plant Genomic DNA Kit (Cat. No. TSP101) (TsingKe Biotech. Co., Ltd., Beijing, China). The primers bt2a and bt2b were used for PCR amplification of the partial $\beta$-tubulin gene (*BenA*) (*Glass & Donaldson, 1995*), primers AD1 and Q1 for the partial calmodulin gene (*CaM*) (*Wang, 2012*), T2 and E2 for the partial DNA-dependent RNA polymerase II second largest subunit gene (*Rpb2*) (*Jiang et al., 2018*), and ITS5 and ITS4 for the nuclear rDNA ITS1-5.8S-ITS2 (ITS) (*White et al., 1990*). PCR reaction was employed in the 20 µL reaction mixture containing 0.5 µL of each primer (10 pM/µL), 1.0 µL of genomic DNA (10 ng/µL), 10 µL of 2 × PCR MasterMix buffer (0.05 u/µL Taq polymerase, 4 mM MgCl2, 0.4 mM dNTPs), and 8 µL of double-distilled water (Tsingke Biotech. Co. Ltd, Beijng, China). Amplification was performed in an AB 2720 thermal cycler (Applied Biosystems, Foster City, California, USA), with the program consisting of 94 °C for 3 min, 34 cycles of 94 °C for 30 s, 50 °C for 30 s, and 72 °C for 30 s, and a last elongation at 72 °C for 5 min. The amplicons were purified and sequenced in two directions with an ABI 3730 DNA analyzer (Applied Biosystems, Foster City, California, USA). Raw sequences were proofread and edited manually with BioEdit 7.0.9 (*Hall, 1999*). The sequences of the five strains generated in this study were deposited in GenBank (**AS3.15900**: ITS = MW946996, *BenA* = MZ004914, *CaM* = MZ004918, *Rpb2* = MW979253; **AS3.16038**: ITS = MW946993, *BenA* = MZ004911; *CaM* = MZ004915; *Rpb2* = MW979252; **AS3.16039**: ITS = MW946994; *BenA* = MZ004912; *CaM* = MZ004916; *Rpb2* = MW979254; **AS3.16040**: ITS = MW946995, *BenA* = MZ004913, *CaM* = MZ004917, *Rpb2* = MW979255; **AS3.16207**: ITS = OM203537, *BenA* = OM220087, *CaM* = OM220088, *Rpb2* = OM220089).

A pilot phylogenetic analysis based on *BenA* sequences of all the members in sect. *Lanata-Divaricata* was launched first, then only the species of sers. *Janthinella* and *Oxalica* were included in the final analyses to identify these five strains. Besides our five strains,

there were 29 species of the two series in the analyses, among which 34 strains were included in the combined *BenA-CaM-Rpb2* mactrix, 36 strains in *CaM* and *Rpb2* mactrices, and 38 strains in *BenA* and ITS mactrices. In addition, three species of the closely-related sections, namely, *P. stolkiae* CBS 315.67 of sect. *Stolkia*, *P. macrosclerotiorum* AS3.5681 of sect. *Gracilenta* and *P. citrinum* CBS 139.45 of sect. *Citrina* were included in the analyses, among which *P. citrinum* was selected as the outgroup. The edited sequences were aligned with MUSCLE implemented in MEGA 6 with the parameters set as default (*Tamura et al., 2013*),  then the aligned sequences were trimmed to generate sequence matrices. The concatenated *BenA-CaM-Rpb2* and the individual *BenA, CaM, Rpb2*, and ITS matrices were analyzed with the Maximum Likelihood (ML) method and subjected to 1000 bootstrap replications, with substitution model and rates among sites set as K2+G+I for the combined *BenA-CaM-Rpb2*, *BenA*, *CaM* and *Rpb2* sequences, and GTR+G for ITS, which were determined by the "find best DNA model" tool of MEGA 6, with the default parameters used for the best model search. Gaps were treated as partial deletion according to *Hall (2013)*. In *BenA-CaM-Rpb2* analysis, each gene region was not treated as separate partitions.

## Nomenclature

The electronic version of this article in Portable Document Format (PDF) will represent a published work according to the International Code of Nomenclature for algae, fungi, and plants, and hence the new names contained in the electronic version are effectively published under that Code from the electronic edition alone. In addition, new names contained in this work have been submitted to MycoBank from where they will be made available to the Global Names Index. The unique MycoBank number can be resolved and the associated information viewed through any standard web browser by appending the MycoBank number contained in this publication to the prefix "http://www.mycobank.org/MycoTaxo.aspx?Link=T{&}Rec=". The online version of this work is archived and available from the following digital repositories: PeerJ, PubMed Central, and CLOCKSS.

# RESULTS

## Phylogenetic analyses

PCR amplification of *BenA*, *CaM*, *Rpb2* and ITS generated ca. 410 bp, 660 bp, 800 bp, and 600 bp amplicons, respectively. The sequence matrices of *BenA*, *CaM*, *Rpb2*, ITS, and the concatenated *BenA-CaM-Rpb2* had 441, 536, 719, 509 and 1693 characters with gaps, respectively. The phylograms inferred from either the combined *BenA-CaM-Rpb2* or the four individual loci readily confirmed the three novelties (Figs. 1–4, Fig. S1).

According to the phylograms, one new species, namely, *P. donggangicum* is in the clade with *P. koreense*, *P. raperi* and *P. yunnansense* of ser. *Janthinella* in *BenA-CaM-Rpb2*, *BenA*, *CaM*, *Rpb2* phylograms with 100%, 98%, 97%, and 100% bootstrap support, respectively, but ITS phylogram does not show any clades in which the new species belongs.

The other two new species, *i.e.*, *P. hepuense* and *P. jiaozhouwanicum* are in the clade with *P. soosanum*, *P. diatomitis* and *P. oxalicum* of ser. *Oxalica* in *BenA-CaM-Rpb2*,

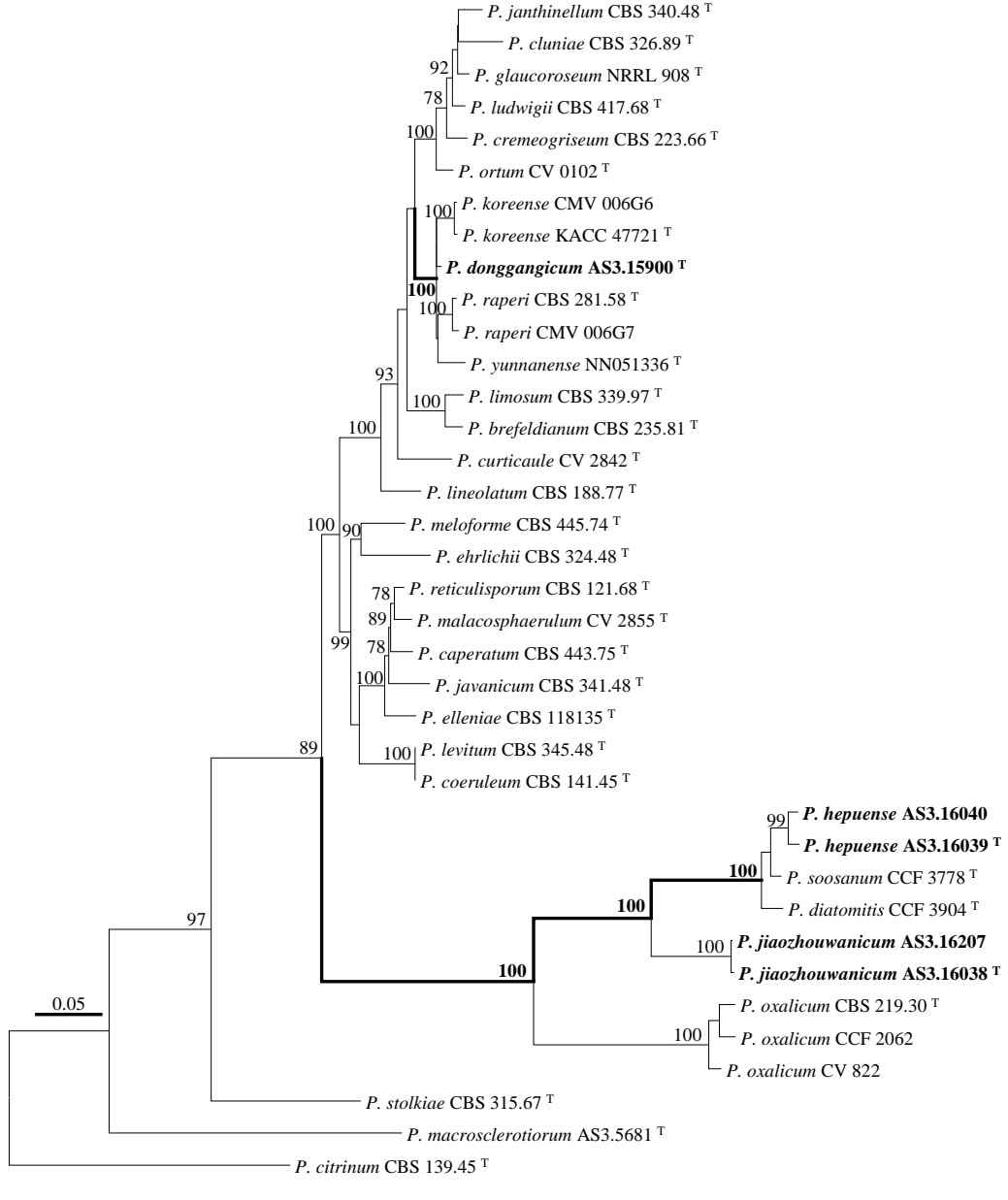

**Figure 1** **ML phylogram inferred from partial *BenA-CaM-Rpb2* sequences.** Bootstrap percentages over 70% derived from 1,000 replicates are indicated at the nodes, [T] indicates ex-type strains, strains belonging to new species are indicated in boldface. Bar = 0.05 substitutions per nucleotide position.

*BenA*, *CaM*, *Rpb2* and ITS phylograms with 100%, 88%, 100%, 96% and 98% bootstrap support, respectively. Among these five species, *P. hepuense*, *P. soosanum* and *P. diatomitis* consistently form an innermost subclade in *BenA-CaM-Rpb2*, *BenA*, *CaM*, *Rpb2* and ITS phylograms with 100%, 100%, 100%, 100% and 79% bootstrap support, respectively, and *P. jiaozhouwanicum* is close-related to them and as an outgroup to this clade with 81%–100% bootstrap support; *P. oxalicum* is as the outermost taxon to these four species.

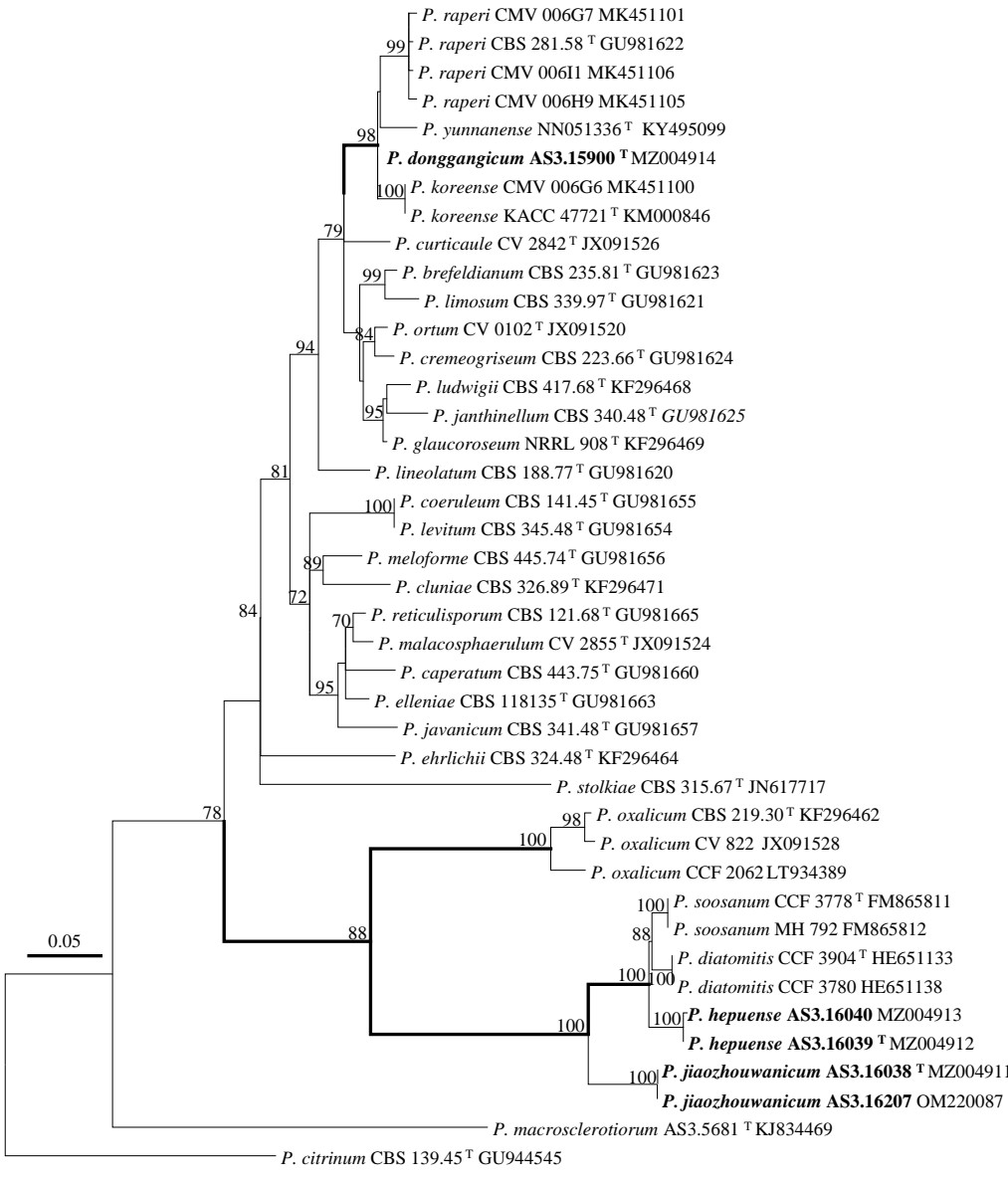

**Figure 2 ML phylogram inferred from partial *BenA* sequences.** Bootstrap percentages over 70% derived from 1,000 replicates are indicated at the nodes, [T] indicates ex-type strains, strains belonging to new species are indicated in boldface. Bar = 0.05 substitutions per nucleotide position.

Although ITS phylogram shows that these five species are in one clade with 98% bootstrap support, the two strains of *P. hepuense* AS3.16039 and AS3.16040 are well separated in different subclades and many clades presented in the protein gene phylograms are absent, thus only the three protein genes has been concatenated for analysis.

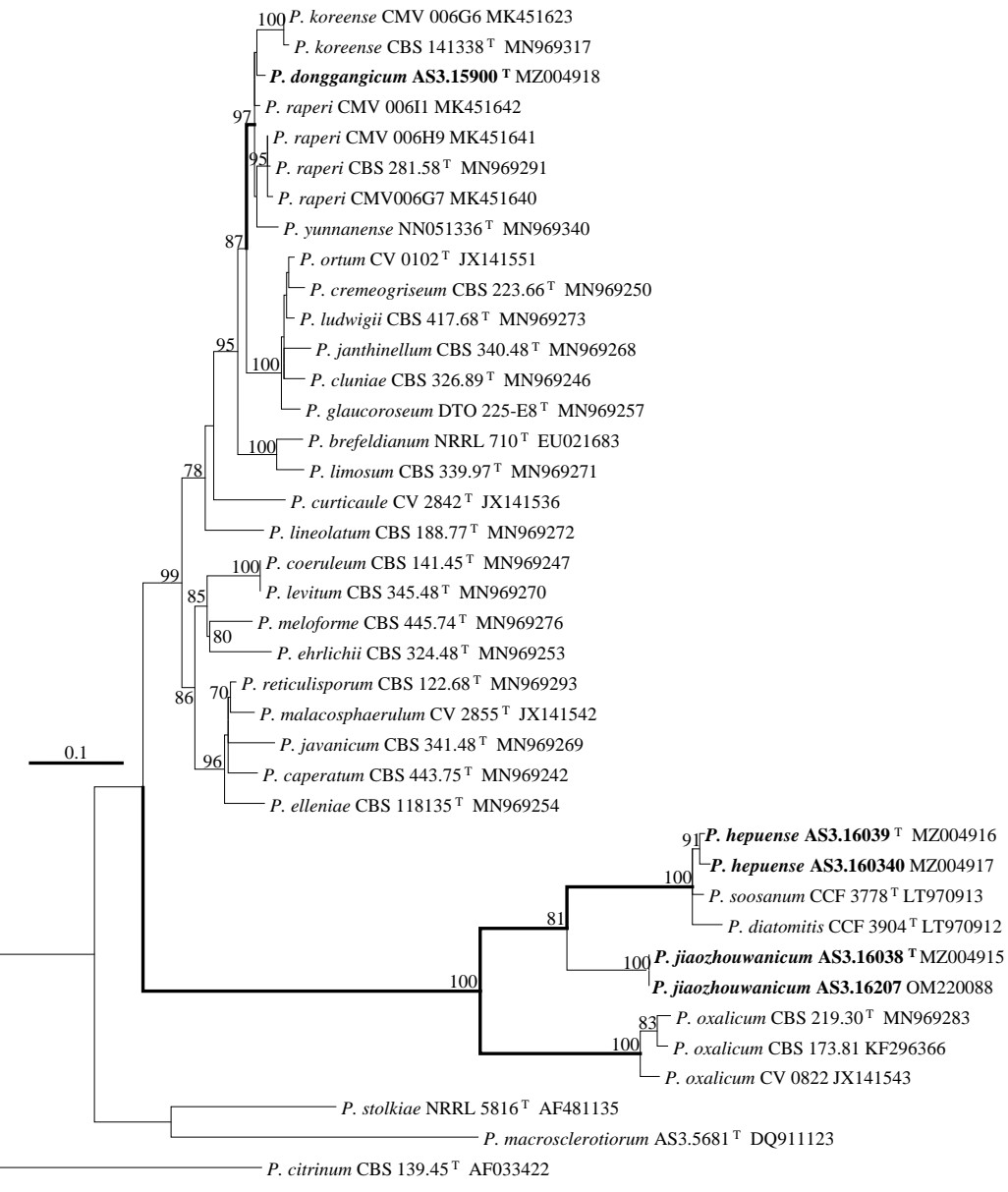

**Figure 3** **ML phylogram inferred from partial *CaM* sequences.** Bootstrap percentages over 70% derived from 1,000 replicates are indicated at the nodes, [T] indicates ex-type strains, strains belonging to new species are indicated in boldface. Bar = 0.1 substitutions per nucleotide position.

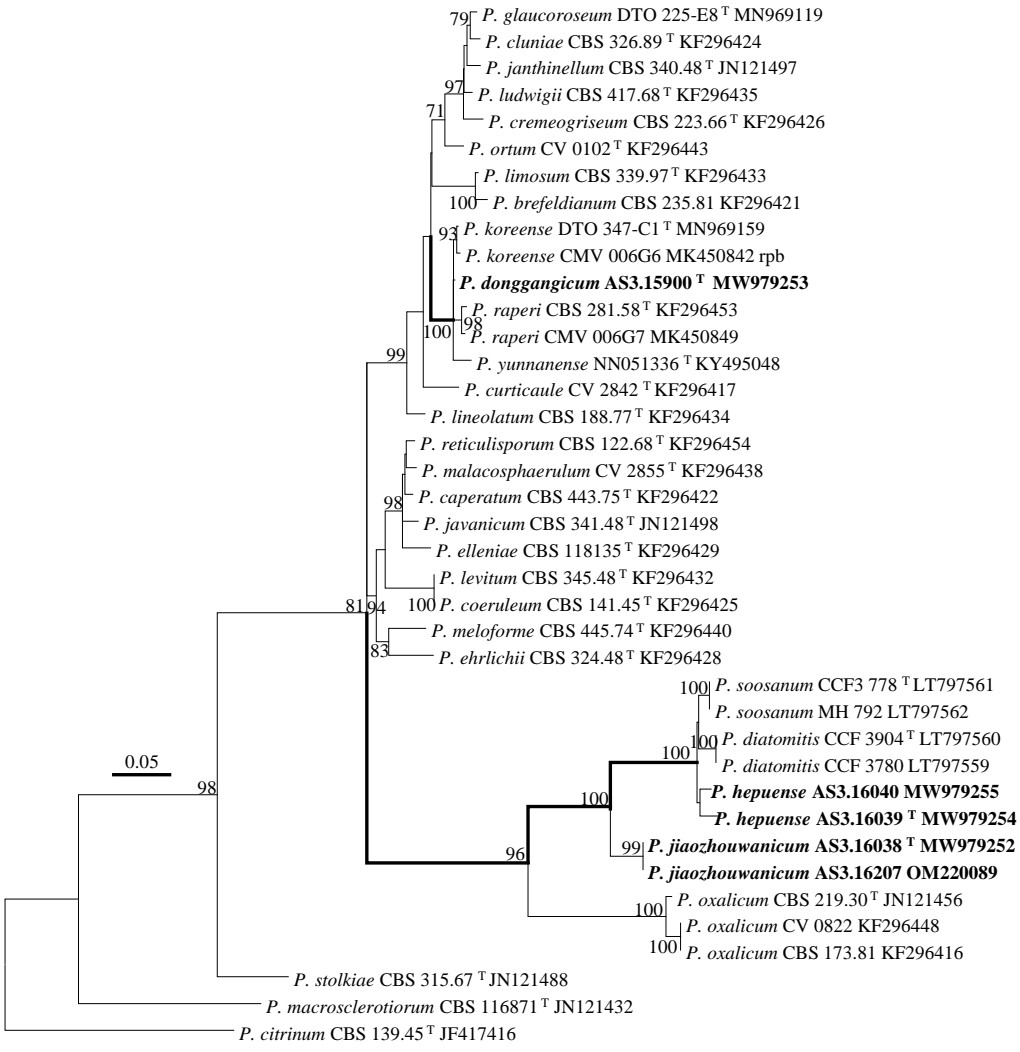

**Figure 4** **ML phylogram inferred from partial *Rpb2* sequences.** Bootstrap percentages over 70% derived from 1,000 replicates are indicated at the nodes, ᵀ indicates ex-type strains, strains belonging to new species are indicated in boldface. Bar = 0.05 substitutions per nucleotide position.

## Taxonomy

***Penicillium donggangicum*** L.Wang, sp. nov.
MycoBank No. MB841518
(Fig. 5)

**Etymology.** The specific epithet is derived from the locale where the ex-type strain was isolated, Donggang, Dandong, Liaoning, China.

**Holotype.** CHINA. LIAONING: Dandong, Donggang, from soil of tidal flats, 39°29′24″N, 124°19′12″E, 39 m, Mar. 29, 2020 (holotype: HMAS350265 from dried

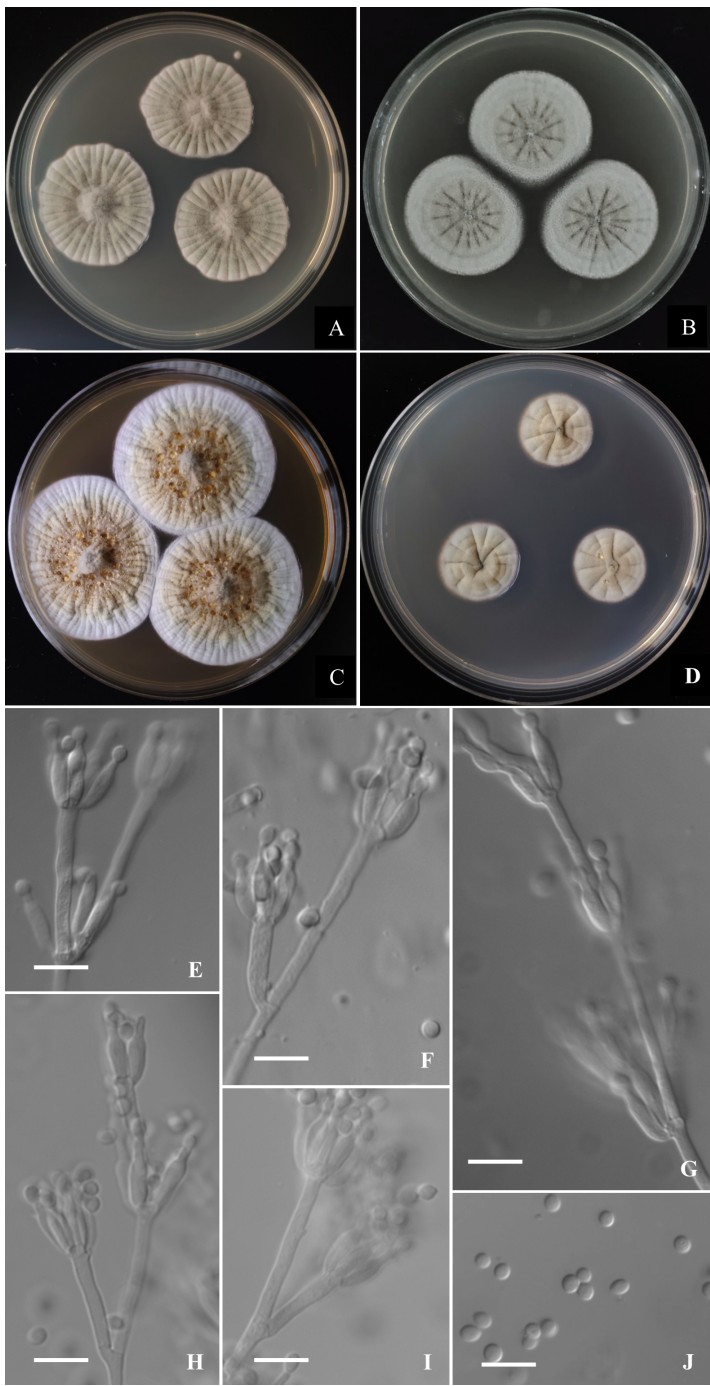

**Figure 5** **Morphological characters of *Penicillium donggangicum* AS3.15900 [T] incubated at 25 °C for 7 days.** (A) CYA. (B) MEA. (C) YES. (D) Cz. (E)–(I) Conidiophores. (J) Conidia. Bar = 10 μm.

culture of ex-type AS3.15900 on CYA), ex-type culture AS3.15900 = LN5H1-4. GenBank: *BenA* = MZ004914, *CaM* = MZ004918, ITS = MW946996, *Rpb2* = MW979253.

**Diagnosis**. This new taxon is characterized by the restricted growth at 37 °C, light yellow sclerotia, sparse sporulation, irregular conidiophores with concurrent phialides and metulae, pyriform to subspherical smooth-walled conidia.

## Description

Colonies 32–34 mm diam on **CYA** at 25 °C after 7 d, thin, radially sulcate densely and centrally raised, margins on agar surface, sinuous, texture velutinous and overlaid with sparse aerial mycelium in central areas; sporulation sparse to moderate, conidia *en masse* near greyish green (27B3–C3); mycelium greyish yellow (1B2–2B2); exudate absent or limited, light yellow; limited sclerotia in central areas, light yellow; soluble pigment absent; moldy odor strong; colony underside near Reddish Blond (5C3). Colonies 30–33 mm diam on **MEA** at 25 °C after 7 d, thin, radially sulcate, margins on agar surface, regular; velutinous and overlaid with sparse mycelium in central areas; sporulation absent to sparse, conidia *en masse* light greyish green; mycelium white at colony margins and pale orange (5A3) in central areas; sclerotia abundant in central areas, light yellow to pale orange (5A3); exudate and soluble pigment absent; moldy odor strong; colony underside near grayish brown (7D3–E4). Colonies 43–45 mm diam on **YES** at 25 °C after 7 d, slightly thick, radially and annularly sulcate densely, centrally raised, margins on agar surface, regular; velutinous; sporulation sparse at the centers, conidia *en masse* grey (3C1); mycelium white at marginal areas and greyish yellow (1B2–2B2) in central areas; light yellow sclerotia abundant in central areas; exudate moderate, light brownish-yellow; soluble pigment absent; moldy odor strong; colony underside near light brown (6D7–D8). Colonies 18–21 mm diam on **CA** at 25 °C after 7 d, thin, radially and annularly plicate, margins submerged, regular; velutinous; sporulation absent or sparse in central areas, light grey; mycelium pale orange (5A2–A3) and white at colony margins; clear exudate absent or limited, light yellow; soluble pigment absent; moldy odor strong; colony underside near grayish brown (7D3–E4). Colonies 12–14 mm diam on CYA at **37 °C** after 7 d, thick, chapped, irregularly sulcate, centrally depressed; velutinous; sporulation sparse, light gray; mycelium dirty-white; exudate and soluble pigment absent; moldy odor strong; colony underside near light yellow. On CYA at **5 °C** after 7 d, no growth. On CREA at 25 °C after 7 d, colonies 16–18 mm diam, acid production absent.

Conidiophores borne from aerial hyphae or surface mycelium, stipes indeterminate, with the diameters about 2.5–3 µm, walls smooth; penicilli irregular, with intercalary phialides and metulae along the conidiophores; metulae 1–3 per verticil, 15–30 × 2–3; phialides, ampulliform, 3–7 per verticil, 9–12 × 2.5–3; conidia pyriform to subspherical, 2.5–3.5 × 2.5–3 µm, smooth-walled.

*Penicillium hepuense* L.Wang, sp. nov.
MycoBank No. MB841525
(Fig. 6)

**Etymology.** The specific epithet is derived from the locale where the ex-type strain was isolated, Hepu, Beihai, Guangxi, China.

**Holotype.** CHINA. GUANGXI: Beihai, Hepu County, from soil of tidal flats, 21°18′13.68″N, 109°23′44.88″E, 10 m, December 30, 2020 (holotype: HMAS350263 from dried culture of ex-type AS3.16039 on CYA), ex-type culture AS3.16039 = TT2-4X3. GenBank: *BenA* = MZ004912, *CaM* = MZ004916, ITS = MW946994, *Rpb2* = MW979254.

**Diagnosis.** This new taxon is characterized by fast growth on CYA and YES, slow growth on MEA, biverticillate and monoverticillate penicilli, ellipsoidal conidia with smooth to finely rough walls.

## Description

Colonies 45–48 mm diam on **CYA** at 25 °C after 7 d, thin, plane, margins on agar surface, regular; velutinous; sporulation heavy, conidia *en masse* near greyish green (25E4); clear exudate limited; soluble pigment absent; colony underside grey, while orange grey (6B2) at the centers. Colonies 26–29 mm diam on **MEA** at 25 °C after 7 d, thin, plane, margins on agar surface, irregular; velutinous, and overlaid with sparse mycelium; sporulation heavy, conidia *en masse* near greyish green to dull green (26E3–E4); exudate and soluble pigment absent; colony underside yellowish grey (M3B2). Colonies 55–59 mm diam on **YES** at 25 °C after 7 d, slightly thick, radially and annularly sulcate, centrally convolute or depressed; velutinous; sporulation moderate to heavy, conidia *en masse* near greyish green (25D3); exudate and soluble pigment absent; colony underside brownish grey (M7C4), and darker at centers. Colonies 27–29 mm diam on **CA** at 25 °C after 7 d, thin, plane, margins submerged, fimbriate; velutinous; sporulation heavy, conidia *en masse* greyish green (25E4); mycelium white at colony margins; exudate and soluble pigment absent; colony underside orange grey (M5B2). On CYA at **37 °C** after 7 d, colonies absent (AS3.16040) or 5–7 mm (AS3.16039), thick, centrally depressed, irregularly plicate; velutinous; sporulation moderate, light gray; exudate and soluble pigment absent; colony underside dark brown. On CYA at **5 °C** after 7 d, no growth. On CREA at 25 °C after 7 d, colonies 20–25 mm diam, acid production absent.

Conidiophores borne from substatum or surface hyphae; stipes (60–) 150–180 (–250) × 3.5–4 μm, smooth-walled; penicilli usually biverticillate, less commonly monoverticillate, metulae 2–4 per verticil, appressed, 15–20 × 3–4 μm; phialides ampuliform, 4–8 per verticil, 12–15 × 3–3.5 μm; conidia ellipsoidal, (4–) 5–6 × 3–4 μm, walls smooth to finely rough.

Additional strains: CHINA. GUANGXI: Hezhou, Babu District, from soil of the river side, 21°30′38″N, 109°39′58″E, 200 m, Jul 12, 2020, AS3.16040 = TT2-6X3. GenBank: *BenA* = MZ004913, *CaM* = MZ004917, ITS = MW946995, *Rpb2* = MW979255.

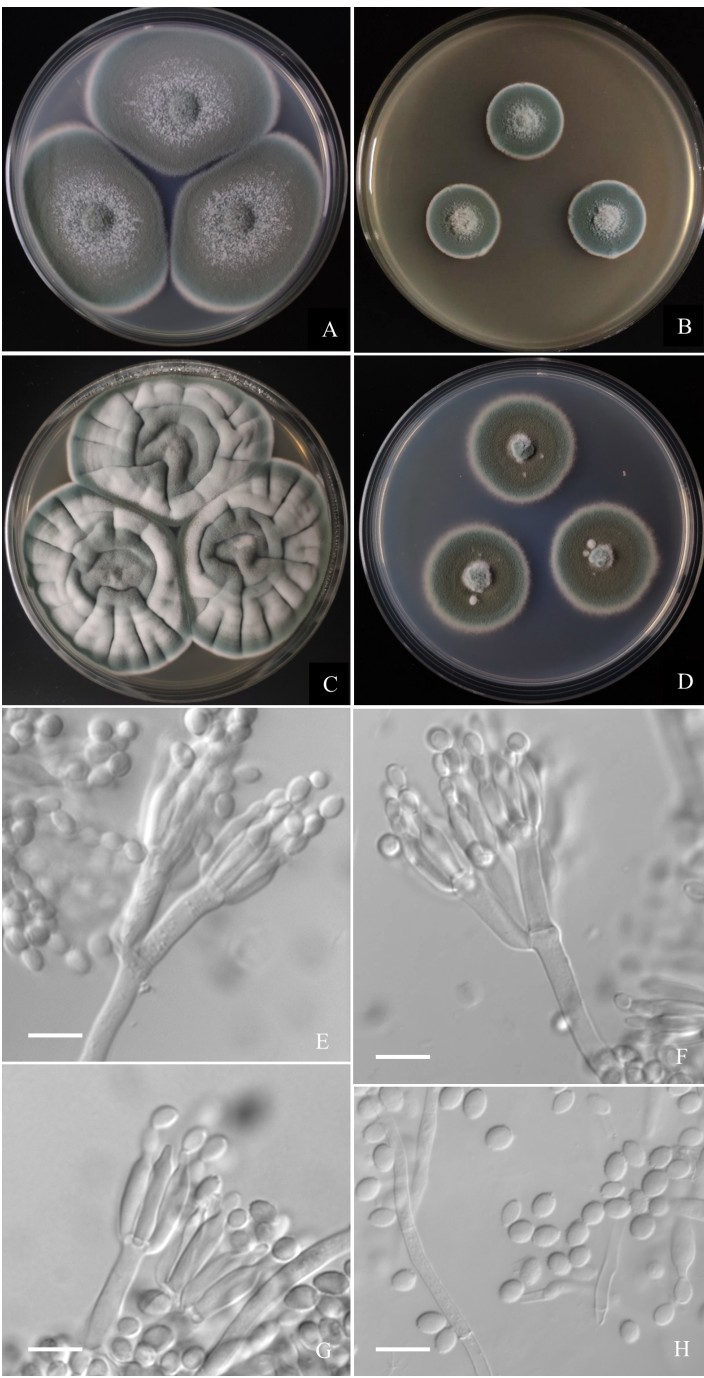

**Figure 6** **Morphological characters of *Penicillium hepuense* AS3.16039[T] incubated at 25 °C for 7 days.**
(A) CYA. (B) MEA. (C) YES. (D) Cz. (E)–(G) Conidiophores. (H) Conidia. Bar = 10 μm.

*Penicillium jiaozhouwanicum* L.Wang, sp. nov.
MycoBank No. MB841531
(Fig. 7)

**Etymology.** The specific epithet is derived from the locale where the ex-type strain was isolated, Jiaozhouwan, Qingdao, Shandong, China.

**Holotype.** CHINA. SHANDONG: Qingdao, Jiaozhouwan, from soil of tidal flats, 37°10′13″N, 120°7′36″E, 3 m, January 29, 2020 (holotype: HMAS350262 from dried culture of ex-type AS3.16038 on CYA), ex-type culture AS3.16038 = 0801H2-2. GenBank: *BenA* = MZ004911, *CaM* = MZ004915, ITS = MW946993, *Rpb2* = MW979252.

**Diagnosis.** This new taxon is characterized by veltutinous colonies and abundant conidia, normal growth at 37 °C, appressed biverticillate penicilli and fusiform smooth-walled conidia.

## Description

Colonies 37–40 mm diam on **CYA** at 25 °C after 7 d, thin, radially sulcate, colony margins on agar surface, regular; velutinous; sporulation abundant, conidia *en masse* greyish green to dull green (27D4–E4), easily falling off; mycelium white at colony margins; exudate and soluble pigment absent; colony underside near greyish green (30B3–B4). Colonies 34–37 mm diam on **MEA** at 25 °C after 7 d, thin, plane, margins on agar surface, regular; velutinous; sporulation abundant, conidia *en masse* near greyish green to dull green (27D4–E4), easily falling off; mycelium white at colony margins; exudate and soluble pigment absent; colony underside near brownish grey (6C2). Colonies 57–60 mm diam on **YES** at 25 °C after 7 d, densely radially to irregularly sulcate with convoluted centers, margins submerged, regular; velutinous; sporulation abundant, conidia *en masse* near greyish green (27C3), easily falling off; mycelium white at colony margins; exudate and soluble pigment absent; colony underside commonly near greyish yellow (3C3) at margins, while dull yellow (3B3) at centers. Colonies 26–29 mm diam on **CA** at 25 °C after 7 d, thin, radially sulcate slightly; margins on agar surface, regular; velutinous; sporulation abundant, conidia *en masse* near greyish green to dull green (27C3–D3), easily falling off; mycelium white at colony margins; exudate and soluble pigment absent; colony underside near greyish green (30C5). Colonies 23–25 mm diam on CYA at **37 °C** after 7 d, thin, radially sulcate, centrally convolute, margins on surface, lightly sinuous; sporulation abundant, conidia *en masse* near greyish green (27C3), easily falling off; mycelium white at colony margins; exudate and soluble pigment absent; colony underside near light brown at margins, while dark brown at centers. On CYA at **5 °C** after 7 d, no growth. On CREA at 25 °C after 7 d, colonies 28–30 mm diam, acid production weak, just underneath the colony.

Conidiophores borne from substratum; stipes from substratum 140–240 × 3–3.5 μm, smooth-walled; penicilli biverticillate, occasionally with one subterminal branch bearing biverticillate or monoverticillate penicilli; metulae 2–4 per verticil, appressed, 13–15 × 3–3.5; phialides, ampuliform with long collula, 4–8 per verticil, 10–13 × 2.5–3.5; conidia fusiform, seldom ellipsoidal, (3.5–) 4–5.5 × 2–2.5 (–3) μm, smooth-walled.

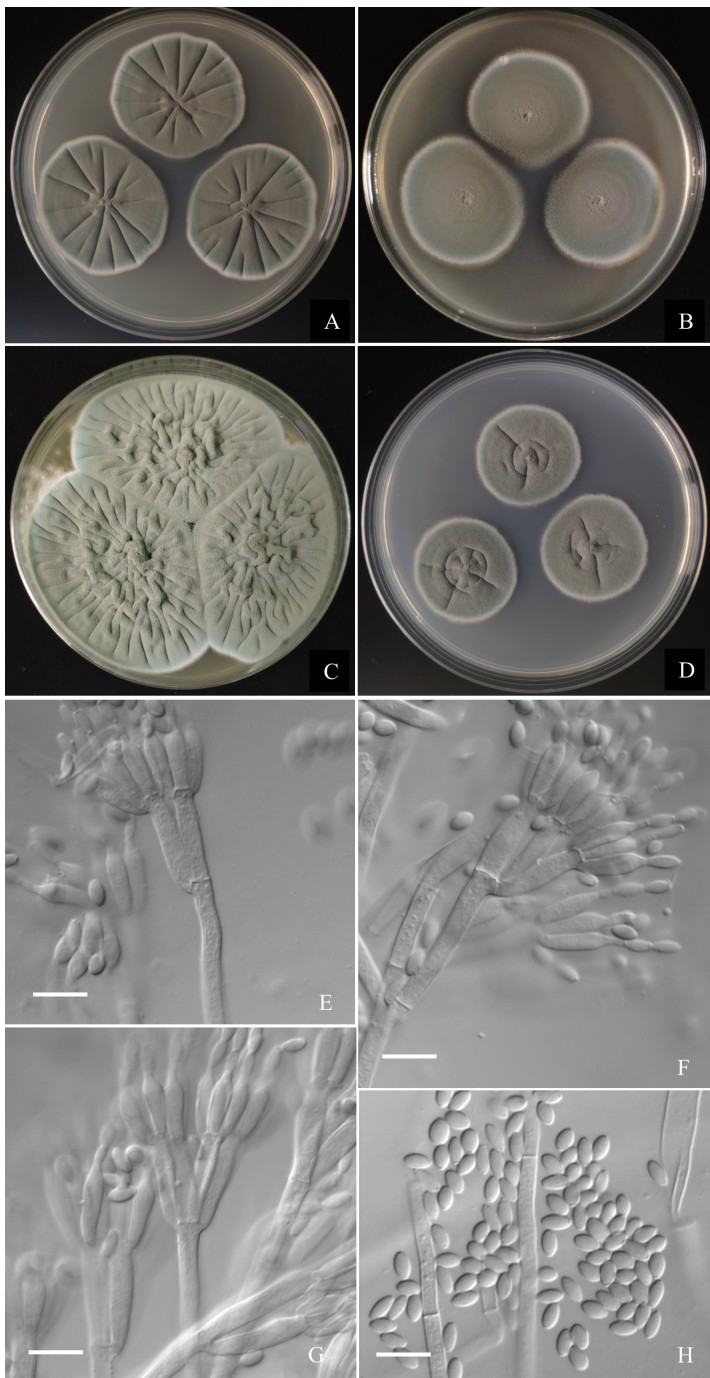

**Figure 7** **Morphological characters of *Penicillium jiaozhouwanicum* AS3.16038^T incubated at 25 °C for 7 days.** (A) CYA. (B) MEA. (C) YES. (D) Cz. (E)–(G) Conidiophores. (H) Conidia. Bar = 10 μm.

Additional strains: CHINA. FUJIAN: Zhangzhou, Jiulongkou, from soil of tidal flats, 24°25′57N, 117°47′28E, 32 m, Apr 22, 2021, AS3.16027 = ZZ2-9-3. GenBank: *BenA* = OM220087, *CaM* = OM220088, ITS = OM203537, *Rpb2* = OM220089.

## DISCUSSION

According to the taxonomic scheme of *Houbraken et al. (2020)*, *P. donggangicum* belongs to ser. *Jathinella*; *P. hepuense* and *P. jiaozhouwanicum* belong to ser. *Oxalica* of sect. *Lanata-Divaricata* in subgen. *Aspergilloides*. In the phylogenies inferred from the three protein genes, *P. donggangicum*, *P. koreense*, *P. raperi* and *P. yunnanense* consistently form a clade with strong bootstrap support, and *P. jiaozhouwanicum*, *P. hepuense*, *P. soosanum*, *P. diatomitis* and *P. oxalicum* are closely related with moderate to strong bootstrap support (Figs. 1–4, Fig. S1).

The morphological characters, such as sparse sporulation, light yellow mycelium and the intercalary vericils of phialides and metulae of *P. donggangicum* quite resemble those of *P. raperi* and the phylogenies based on the protein-coding genes also showed the close relationship between them. But physiologically, *P. donggangicum* grows faster than *P. raperi* at 25 °C (CYA: 32–34 mm *vs.* 21–23 mm, MEA: 30–33 mm *vs.* 23–24 mm, YES: 43–45 mm *vs.* 27–28 mm) and the colony reverse on CYA of *P. donggangicum* shows orange-brown while that of *P. raperi* shows olive color. Moreover, *P. donggangicum* produces abundant sclerotia, while *P. raperi* does not produce sclerotia. Though the conidiophores of the two species are both from aerial hyphae with intercalary metulae and phialdes, and their metulae are unequal in length and divergently positioned, the diameters of the conidiophore elements of the new species are relatively larger than those of *P. raperi* (2–3 μm *vs.* 1.3–2.2 μm) (*Smith, 1957*), and the conidia of *P. donggangicum* are smooth-walled but those of *P. raperi* are slightly rough (*Smith, 1957*; *Visagie et al., 2015*). *P. donggangicum* shows faster growth than *P. yunnanense* at 25 °C on CYA (32–34 mm *vs.* 23–25 mm) and YES (43–45 mm *vs.* 22–23 mm), in addition, *P. donggangicum* produces abundant sclerotia, light yellow mycelium and grows at 37 °C, but *P. yunnanense* does not produce sclerotia, bears white mycelium and cannot grow at 37 °C. Microscopically, *P. donggangicum* bears concurrent verticils of phialides and metulae along the trailing conidiophores, whereas *P. yunnanense* does not have this character (*Diao et al., 2018*). The sparse sporulation, abundant sclerotia, light yellow mycelium and concurrent phialides and metulae also distinguish *P. donggangicum* from *P. koreense*, which has moderate to heavy sporulation, white mycelium and predominantly monoverticillate penicilli, and lacks sclerotia (*You et al., 2014*). A synopsis of these morphological comparisons among the four species is given in Table 1.

Only three species were included in ser. *Oxalica* before this study (*Houbraken et al., 2020*). There are many characters shared by the taxa in this series, such as good growth at 25 °C and 37 °C, typical velutinous colonies with abundant dark green conidia *en masse*, predominantly biverticillate penicilli with 2–4 appressed metulae, and ellipsoidal conidia. One new member introduced to this series here, namely, *P. hepuense* possesses most of these common characters but its maximal growth rate at 25 °C is much less than those

**Table 1 Comparisons of morphological characters among *P. donggangicum*, *P. rperi*, *P. yunnanense* and *P. koreense*.**

| | *P. donggangicum* | *P. raperi* [a] | *P. yunnanense* [b] | *P. koreense* [c] |
|---|---|---|---|---|
| On CYA at 25 C, 7 d diam (mm) | 32–34 | 21–23 | 23–25 | 32–34 |
| On MEA at 25 C, 7 d diam (mm) | 30–33 | 23–24 | 33–34 | 42–48 |
| On YES at 25 C, 7 d diam (mm) | 43–45 | 27–28 | 22–23 | N/A |
| On CYA at 37 C, 7 d diam (mm) | 12–14 | 15–16 | Absent | 15–19 |
| Sclerotia | Abundant | Absent | Absent | Absent |
| Mycelium color | Yellow | Pale yellow to mauve | White | White |
| Sporulation | Sparse | Sparse | Sparse to moderate | Moderate to heavy |
| Penicilli | Irregular, with intercalary metulae and phialides | Irregular, with intercalary metulae and phialides | Monoverticillate to irregular | Predominantly monoverticillate |
| Conidiophore elements diam (μm) | 2–3 | 1.3–2.2 | 2–4 | 2–3 |
| Conidia | Pyriform to subspherical, smooth | Irregularly ovate, slightly rough | Broadly ellipsoidal to subglobose, smooth | Globose to broadly ellipsoidal, smooth or finely rough |

**Notes.**
[a]Data from *Smith (1957)* and *Visagie et al. (2015)*.
[b]Data from *Diao et al. (2018)*.
[c]Data from *You et al. (2014)*.

of the other members (29 mm *vs.* 62 mm for *P. oxalicum*, 62 mm for *P. diatomitis*, 57 mm for *P. soosanum*), and it does not grow at 37 °C, or only forms colonies about 5–7 mm diam, while the other members all grow well at 37 °C. It can be further distinguished from *P. oxalicum* by the absence of acid production on CREA, from *P. diatomitis* by longer conidia (4–6 μm *vs.* 3.5–4.5 μm) and producing a portion of monoverticillate penicilli, from *P. soosanum* by the smooth to finely rough ellipsoidal conidia and the production of a portion of monoverticillate penicilli. Another new species, *i.e.*, *P. jiaozhouwanicum* is different from the above four species by its distinctive smooth-walled fusiform conidia, and metulae of nearly equal length. In pairwise comparison, *P. jiaozhouwanicum* differs from *P. oxalicum* by the weaker acid production on CREA ("underneath the colony" *vs.* "in colony periphery"), and much shorter stipes in maximal length (240 μm *vs.* 550 μm) (*Kubátová et al., 2019*). Though the isolate IJFM 2062 (ex-type of "*P. asturianum*") also produces conidia in fusiform, their walls are roughened and the roughening is arranged in spiral bands (*Ramírez & Martínez, 1981*). The lower growth rate on CYA and MEA at 25 °C and faster growth at 37 °C of *P. jiaozhouwanicum* allow its distinction from *P. diatomitis* (CYA: 37–40 mm *vs.* 45–65 mm, and MEA: 34–37 mm *vs.* 48–62 mm at 25 °C; 20–23 mm *vs.* 7–14 mm on CYA at 37 °C). *P. jiaozhouwanicum* can be further distinguished from *P. soosanum*

**Table 2  Comparisons of morphological characters among *P. jiaozhouwanicum*, *P. hepuense*, *P. oxalicum*, *P. diatomitis* and *P. soosanum*.**

| | *P. hepuense* | *P. jiaozhouwanicum* | *P. oxalicum* [a] | *P. diatomitis* [a] | *P. soosanu* [a] |
|---|---|---|---|---|---|
| On CYA at 25 C, 7 d diam (mm) | 45–48 | 37–40 | 36–62 | 45–65 | 45–65 |
| On MEA at 25 C, 7d diam (mm) | 26–29 | 34–37 | 25–62 | 48–62 | 55–65 |
| On YES at 25 C, 7d diam (mm) | 55–59 | 57–60 | 37–63 | 59–67 | 60–69 |
| On CYA at 37 C, 7 d diam (mm) | 0–7 | 23–25 | 10–38 | 5–25 | 5–20 |
| On CREA at 25 C, 7 d diam (mm); acid production | 20–25; absent | 28–30; weak | 12–19; poor | 7–13; absent | 13–17; absent |
| Penicilli | Biverticillate, rarely monoverticillate | Biverticillate, rarely with a subterminal metula | Biverticillate and monoverticillate, rarely terverticillate | Biverticillate | Biverticillate, rarely with a subterminal metula |
| Stipe length (μm) | 60–250 | 140–240 | 75–550 | 75–300 | 60–250 |
| Conidia | Ellipsoidal, smooth to finely rough | Fusiform, seldom ellipsoidal, smooth | Ellipsoidal, finely rough | Ellipsoidal, finely rough | Broadly ellipsoidal, subglobose, rough |

**Notes.**
[a] Data from *Kubátová et al. (2019)*.

by producing smooth-walled, fusiform conidia, while the latter has roughened, subglobose to broadly ellipsoidal conida. In addition, *P. jiaozhouwanicum* shows slower growth at 25 °C (CYA: 37–40 mm *vs.* 45–65 mm, MEA: 34–37 mm *vs.* 55–65 mm) but faster growth at 37 °C (20–23 mm *vs.* 5–20 mm) than *P. soosanum* (Kubátová et al., 2018). Comparing to *P. hepuense*, *P. jiaozhouwanicum* grows faster on MEA at 25 °C (34–37 mm *vs.* 26–29 mm) and grows moderately at 37 °C (20–23 mm *vs.* 0–7 mm), *P. hepuense* bears a portion of monoverticillate penicilli, but *P. jiaozhouwanicum* commonly has biverticillate ones. A synopsis of these morphological comparisons among these five species is given in Table 2.

In this study, we employed a culture-dependent approach to isolate the culturable fungi and identified them using a polyphasic taxonomic method aiming to enrich the culture collection from various environmemts. Though the NGS using Illumina sequencing platform is widely used in the metabarcoding studies on environmental mycobiota, flaws exist in each step of the workflow (*e.g.*, *Hugerth & Andersson, 2017*). Particularly, the genetic markers commonly used are the sub-regions of ITS1-5.8S-ITS2, namely, ITS1 and ITS2, which are as short as about only 360 bp and 300 bp, respectively (*e.g.*, *Francioli et al., 2021*). According to literatures and our experience, ITS1-5.8S-ITS2 alone cannot provide the accurate identification of species of certain widely distributed genera, such as *Aspergillus*, *Cladosporium*, *Fusarium*, *Penicillium*, *Talaromyces*, etc. Some species in *Aspergillus* and *Penicillium* have identical ITS sequences, for instance, *A. flavus* and *A. minisclerotigenes*; *P. echinulatum* and *P. solitum*, etc. (*e.g.*, *Houbraken, Visagie & Frisvad, 2021*). Although certain protein genes can be used as the alternative barcodes in NGS, they are usually genus-specific and not easy to be PCR-amplified, and their sequence databases are incomplete. Therefore, the ployphasic method including phenotypic, phylogenetic

and metabarcoding approaches in studying environmental mycobiota is recommended, which would present both the outlines of species richness and the accuracy and precision of species delimitation if properly used (*e.g.*, *Lücking et al., 2020*).

### Funding
This work was supported by the National Natural Science Foundation of China (U20A20101) and the National Project on Scientific Groundwork, Ministry of Science and Technology of China (2019FY100700). The funders had no role in study design, data collection and analysis, decision to publish, or preparation of the manuscript.

### Grant Disclosures
The following grant information was disclosed by the authors:
National Natural Science Foundation of China: U20A20101.
National Project on Scientific Groundwork, Ministry of Science and Technology of China: 2019FY100700.

### Competing Interests
The authors declare there are no competing interests.

### Author Contributions
- Ke-Xin Xu performed the experiments, analyzed the data, prepared figures and/or tables, and approved the final draft.
- Xia-Nan Shan performed the experiments, prepared figures and/or tables, and approved the final draft.
- Yongming Ruan analyzed the data, prepared figures and/or tables, authored or reviewed drafts of the paper, and approved the final draft.
- JianXin Deng conceived and designed the experiments, authored or reviewed drafts of the paper, and approved the final draft.
- Long Wang conceived and designed the experiments, analyzed the data, authored or reviewed drafts of the paper, and approved the final draft.

### Data Availability
The phylograms based on ITS and seqence matrices of BenA-CaM-Rpb2, BenA, CaM, Rpb2 and ITS are available in the Supplementary Files.

### New Species Registration
The following information was supplied regarding the registration of a newly described species:
*Penicillium donggangicum* L.Wang, sp. nov. MycoBank: MB841518.
Penicillium hepuense L.Wang, sp. nov. MycoBank: MB841525.
Penicillium jiaozhouwanicum L.Wang, sp. nov. MycoBank: MB841531.

## Supplemental Information

Supplemental information for this article can be found online at http://dx.doi.org/10.7717/peerj.13224#supplemental-information.

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
