# Peer review of "Three new Penicillium species isolated from the tidal flats of China"

_PeerJ, doi:10.7717/peerj.13224_

## Round 0.1 · original submission · Major Revisions

Three experts in this field assessed your manuscript and found major flaws in both the study design and results presentation.

Please address all the comments that are part of this letter.

Reviewer 1 ·

Basic reporting

The authors present the identification of three new species of Penicillium isolated from tidal flats of China.
The manuscript is mostly well written in professional English and in an ordered manner that makes it easy to follow. Some grammatical errors are found that should be corrected before publication. For example, (errors marked between parentheses):

Lines 44-46. Either a word is missing in the first line ("...coastal environments ARE based on both...") or the phrase is incomplete/truncated at the end.

Lines 46-47. Singular/plurals are mixed: "...carried out an extensive studies..." should be either "...carried out extensive studies..." or "...carried out an extensive study...".

These are just two examples but there are others. The full manuscript should be thoroughly checked to detect any other grammatical errors before it is ready for publication.

To the best of my knowledge, the appropriate literature has been cited.

Regarding the overall presentation of the manuscript, my only other concern is the final presentation of contrasting/identifying characteristics of the new species. A table summarizing what is explained in the discussion section would be of great aid. Also, highlighting the size bar for figure 1 (maybe changing to bold) would also be beneficial (it is hard to find as it is).

Experimental design

The experimental design of the isolation, molecular identification, and morphological characterization are well planned but some extra clarifications should be included.

Lines 91-95. The names of the primers used for each case should be included (it could be a table). For example, White et al. (1990) report 5 primers for amplification of the spacer region. It is not clear which pair of these 5 primers were used since two combinations are able to amplify the full region (i.e., ITS5-ITS4 or ITS1-ITS4).

Lines 95-96. To help make the manuscript self-contained please add the Tm used for each set of primers (this could be included in the proposed table).

Lines 111-112. Authors state that "sequences were aligned with MUSCLE or ClusalW implemented in MEGA6". For which sequences/regions was MUSCLE used and for which cases was ClustalW used? Also, the used parameters for each one should be noted (if default parameters were used this should be explicitly stated).

Lines 115-116. How were the substitution models and rates among sites chosen? If the "find best DNA model" tool of MEGA was used this should be stated. Also, the parameters used for the best model search should be included.

Validity of the findings

Regarding the analysis of the phylogenetic results, I have one concern about the ITS region.

Authors state that the ITS phylogram "does not show any clades the new species belong to" (lines 140-141) and that "Although the ITS phylogram also supports this clade, the two strains of P. hepuense AS3.16039 and AS3.16040 are separated, so only the three protein genes were used in the analysis of concatenated loci (Fig. 1, Figs S1–4).". The amplicons for the ITS regions are 800 bp long (lines 132-133) but once aligned the remaining sequences after trimming are quite short (363 nt). After reviewing their supplemental data I noticed that the sequences contain almost only the 5.8S region. Most of the ITS1 and ITS2 regions are truncated. Since the 5.8S region is very conserved and slowly evolving, only a handful of sites are phylogenetically informative, and not enough resolving power is available.
If the ends of the sequences were trimmed due to poor alignments, a different algorithm should be tried. ClustalW and MUSCLE have been shown to perform poorly for ITS regions (https://pubmed.ncbi.nlm.nih.gov/23185439/). I recommend using instead MAFFT with the E-INS-i option (https://mafft.cbrc.jp/alignment/server/).
If, on the other hand, the ends were trimmed due to incomplete sequencing or bad quality at the extremes this should be stated.
Although the other marker regions provide good support for the placement of the three new species, care should be taken to correctly try to make the best use of the ITS sequences or point out its incompleteness.

Reviewer 2 ·

Basic reporting

This paper is reporting three new species of Penicillium. The methodology is sound and the descriptions and illustrations are OK. I only criticise that of the two new species there is only one strains so more detailed comparison between other taxa and variability within the taxon is not possible. I also notice that the strains are not deposited in recognized culture collection so the specimens or cultures are not available to the international community. The referecnes to the literature should be updated. Some articles and books have recently been published.

Experimental design

Experimental design is OK

Validity of the findings

The report of the newly described species is OK

Reviewer 3 ·

Basic reporting

* Grammar needs a lot of work. There are also many names spelled incorrectly, e.g. Gibberella.
* Don’t believe that it is good enough to only say that ‘Genomic DNA extraction was performed according to Wang and Zhuang (2004)’. Unless Wang and Zhuang designed a new extraction method, please add details of the method used. The same for the PCR conditions.

Experimental design

* Primer information is missing, e.g. what primers were used? Glass and Donaldson 1995 have more than one primer combination for BenA.
* Based on the text, it is difficult to follow what phylogenies were calculated.

Validity of the findings

* My biggest problem with the manuscript in this form, is the fact that sequences of the proposed new species were compared with only ex-types of close relatives. This is in my opinion a bad approach as we don’t know what the infra species variation is like in these clades. In both cases, there are more sequences available on GenBank that can be included in the phylogenies.

Additional comments

In the introduction:
* Please make sure to check the numbers presented in Hawksworth and Lucking. It is not 12000 species described.
* Please expand why NGS cannot perform accurate species identifications.
* Add how many species of Penicillium is known. Also, why would we expect there to be many more undiscovered species?

---

## Round 0.2 · Minor Revisions

Two of the original Reviewers assessed the revised version of the manuscript and found it improved. However, one of the Reviewers thinks there is still room for optimization and I do agree. Please address those observations.

Reviewer 1 ·

Basic reporting

The authors have addressed all my concerns and suggestions.

Experimental design

The authors have addressed all my concerns and suggestions.

Validity of the findings

The authors have addressed all my concerns and suggestions.

Reviewer 3 ·

Basic reporting

* The authors made several good additions, however, some of my original concerns remain. The paper is difficult to read and still contains many spelling and grammatical errors that must be attended to before publishing.
* Gibberella is not a genus name used for many years since the adoption of the One Fungus One Name concept. Please use the correct genus name.
* For primers: “primers AD1, AD2 and Q1, Q2 for the partial calmodulin gene (CaM)” and “T1, T2 and E1, E2 for the partial DNA-dependent RNA polymerase II second largest subunit gene (Rpb2)”. Which primers were actually used?
* For the concatenated phylogeny, the authors did not specify whether each gene region was treated as separate partitions.
* Keeping to the phylogenies, please keep the single gene trees also in the main paper. No need to have them as supplementary. Also, for these, why are the good reference strains that is available not included in especially the single gene trees? There are several more strains available for closely related species that have both BenA and RPB2. I include here the other reference sequences that are available
* Aspergillus_yunnanensis_CGMCC3.19711T__ITS_MN066373_BenA_MN072909_CaM_MN072911_RPB2_MN072913
* Aspergillus_yunnanensis_CGMCC3.19712__ITS_MN066374_BenA_MN072910_CaM_MN072912_RPB2_MN072914
* Penicillium_yunnanense_NN051336-1__ITS_KY494989_BenA_KY495098_CaM_KY494929_RPB2_
* Penicillium_yunnanense_CBS144485T__ITS_KY494990_BenA_KY495099_CaM_MN969340_RPB2_KY495048
* Penicillium_soosanum_CCF3905__ITS_FJ430746_BenA_FM865812_CaM__RPB2_LT797562
* Penicillium_soosanum_CCF3776__ITS_LT797554_BenA_HE651132_CaM__RPB2_LT797563
* Penicillium_soosanum_CCF3778T__ITS_FJ430745_BenA_FM865811_CaM_LT970913_RPB2_LT797561
* Aspergillus_ivoriensis_CBS551.77T__ITS_EF652441_BenA_EF652265_CaM_EF652353_RPB2_EF652177
* Aspergillus_raperi_NRRL2640__ITS_EF652453_BenA_EF652277_CaM_EF652365_RPB2_EF652189
* Aspergillus_raperi_CBS123.56T__ITS_EF652454_BenA_EF652278_CaM_EF652366_RPB2_EF652190
* Aspergillus_raperi_NRRL5039__ITS_EF652496_BenA_EF652320_CaM_EF652408_RPB2_EF652232
* Penicillium_raperi_PPRI9606_CMV006H9__ITS_MK450712_BenA_MK451105_CaM_MK451641_RPB2_MK450850
* Penicillium_raperi_PPRI25781_CMV006G7__ITS_MK450711_BenA_MK451101_CaM_MK451640_RPB2_MK450849
* Penicillium_raperi_PPRI9609_CMV006I2__ITS_MK450714_BenA_MK451107_CaM_MK451643_RPB2_
* Penicillium_raperi_PPRI9611_CMV006I1__ITS_MK450713_BenA_MK451106_CaM_MK451642_RPB2_
* Penicillium_raperi_PPRI25889_CMV012C2__ITS_MK450715_BenA_MK451243_CaM_MK451644_RPB2_
* Penicillium_raperi_CBS281.58T__ITS_AF033433_BenA_GU981622_CaM_MN969291_RPB2_KF296453
* Penicillium_diatomitis_CCF3906__ITS_FJ430747_BenA_HE651136_CaM__RPB2_LT797557
* Penicillium_diatomitis_CCF3779__ITS_HE651147_BenA_HE651137_CaM__RPB2_LT797558
* Penicillium_diatomitis_CCF3780__ITS_HE651148_BenA_HE651138_CaM__RPB2_LT797559
* Penicillium_diatomitis_CCF4379__ITS_HE651151_BenA_HE651134_CaM__RPB2_LT797556
* Penicillium_diatomitis_CCF3907__ITS_FJ430749_BenA_FM865813_CaM__RPB2_LT797555
* Penicillium_diatomitis_MH248__ITS_HE651150_BenA_HE651135_CaM__RPB2_
* Penicillium_diatomitis_MH285__ITS_HE651149_BenA_HE651139_CaM__RPB2_
* Penicillium_diatomitis_CCF3904T__ITS_FJ430748_BenA_HE651133_CaM_LT970912_RPB2_LT797560
* Penicillium_koreense_PPRI25780_CMV006G6__ITS_MK450699_BenA_MK451100_CaM_MK451623_RPB2_MK450842
* Penicillium_koreense_KACC46682__ITS_KM048199_BenA_KM000844_CaM__RPB2_
* Penicillium_koreense_KACC47720__ITS_KM048200_BenA_KM000845_CaM__RPB2_
* Penicillium_koreense_KACC47722__ITS_KM048201_BenA_KM000847_CaM__RPB2_
* Penicillium_koreense_KACC47721T__ITS_KJ801939_BenA_KM000846_CaM_MN969317_RPB2_MN969159
* The diagnoses provided for the new species do not actually distinguish them from closely related species. Can the authors rather focus on discussing the morphological differences between these species?
* Morphological comparisons of new species to close relatives: Please cite appropriate references. e.g. From the text, it is not clear who reported P. raperi as having ‘conidiophore elements’ sized 1.3-2.2? The use of ‘fast growth’ is not appropriate and only relative. I prefer the use of ‘faster growth than’.
* “…it grows more slowly on MEA at 25 °C than the other members (26–29 mm vs. 25–62 mm for P. oxalicum”. Growth thus still in the range of P. oxalicum? Similarly, “on shorter stipes (120–240 μm vs. 75–550 μm).” these ranges overlaps? Please rephrase where these types of statements are made
* I would be careful of these types of statements “Though the NGS using Illumina sequencing platform was widely used in the metabarcoding studies on environmental mycobiota, flaws exist in each step of the workflow.” without providing citations or what is meant. Metabarcoding is incredibly powerful to study diversity.

Experimental design

See above

Validity of the findings

I believe the species are new, but the data, analyses and comparisons are not up to standard. See comments above.

---

## Round 0.3 · accepted · Accept

Thanks for the modifications to the text, the manuscript is now suitable for publication in PeerJ.